# Energy and Carbon Considerations of Fine-Tuning BERT

**Xiaorong Wang**[1*]
swangxr@gmail.com

**Clara Na**[2*]
csna@cs.cmu.edu

**Emma Strubell**[2,3]
strubell@cmu.edu

**Sorelle A. Friedler**[1]
sorelle@cs.haverford.edu

**Sasha Luccioni**[4]
sasha.luccioni@hf.co

[1]Haverford College, [2]Carnegie Mellon University, [3]Allen Institute for AI, [4]Hugging Face

## Abstract

Despite the popularity of the pre-train then fine-tune paradigm in the NLP community, existing work quantifying energy costs and associated carbon emissions has largely focused on language model pre-training. Although a single pre-training run draws substantially more energy than fine-tuning, fine-tuning is performed more frequently by many more individual actors, and thus must be accounted for when considering the energy and carbon footprint of NLP. In order to better characterize the role of fine-tuning in the landscape of energy and carbon emissions in NLP, we perform a careful empirical study of the computational costs of fine-tuning across tasks, datasets, hardware infrastructure and measurement modalities. Our experimental results allow us to place fine-tuning energy and carbon costs into perspective with respect to pre-training and inference, and outline recommendations to NLP researchers and practitioners who wish to improve their fine-tuning energy efficiency.

## 1 Introduction

Fine-tuning pre-trained language models is a frequent occurrence in natural language processing (NLP) research and practice, yet the vast majority of work quantifying the energy and carbon footprints of NLP workloads has focused on pre-training (Strubell et al., 2019; Dodge et al., 2022; Luccioni et al., 2023) or inference (Desislavov et al., 2023; Luccioni et al., 2023). The typical lifecycle of an NLP model includes data ingestion, pre-training, fine-tuning and inference, all of which contribute non-trivially to energy use and corresponding emissions (Patterson et al., 2021; Wu et al., 2022). Better understanding of the role that each phase plays in overall energy and carbon footprint is vital to inform policy decisions, yet we still lack basic data quantifying the rela-

tive contributions due to different aspects of model development and use (Kaack et al., 2022).

In this work we perform an empirical study to quantify the energy requirements of language model fine-tuning, including in the context of pre-training and inference energy requirements. While this may seem like it should be a straightforward calculation, there are several variables that can influence compute time and energy consumption, ranging from: (1) the type of hardware used for both pre-training and fine-tuning (since this usually differs between the two), (2) the type of task and the type of computation required to carry it out, and (3) intrinsic characteristics of the dataset, such as average sequence length, its similarity with the pre-training dataset, etc.

In order to isolate the factors that have the most influence on fine-tuning dynamics, we compare fine-tuning energy use across a suite of common supervised NLP datasets including the tasks of entailment, sentiment analysis, question answering, and named entity recognition, and with training data sizes ranging from 6K to 400K examples. We also measure energy use across two different sets of hardware, using the CodeCarbon (Schmidt et al., 2021) software package and a physical energy measurement device at the wall, to quantify variance due to physical factors. To enable carefully controlled comparison of the roles of pre-training and fine-tuning in NLP model lifecycle energy use, we additionally pre-train BERT variants from scratch on the same hardware. We find that pre-training BERT is equivalent to anywhere from 400 (MNLI) to 45,000 (RTE) fine-tuning runs depending on the dataset size, and that number of training tokens[1] is a reasonable heuristic for estimating fine-tuning energy use. Further comparison of fine-tuning in-

---

*Denotes equal contribution.

[1]The "true" number of training tokens seen, accounting for dynamic padding of sequences to the maximum length in a batch, is a better predictor than relying on to mean or median number of tokens per example.

ference energy intensity across tasks confirms that example sequence length holds a much stronger influence on energy intensity in the fine-tuning phase than in the inference phase, in alignment with expectations from previous work (Zhou et al., 2021). Together, our observations contextualize the energy and carbon requirements of fine-tuning in the broader model lifecycle and highlight the need to study fine-tuning energy efficiency separately from pre-training and inference workloads in NLP models. We hope that our careful measurement of the relative costs of different NLP workloads will serve as a valuable datapoint informing decision-making both within and beyond the NLP community.

## 2 Related Work

Measurement of energy consumption and carbon emissions of NLP models has become an active area of research since it was first identified that modern state-of-the-art models based on deep learning can produce substantial greenhouse gas emissions due to the energy required to train them (Strubell et al., 2019; Schwartz et al., 2020). These measurements have mostly focused on two research directions. First, there has been a series of empirical studies on different model architectures, focused on estimating the carbon emissions generated by their training process and the relative efficiency of different methods (Naidu et al., 2021; Patterson et al., 2021, 2022). Recent work by Luccioni et al. has built upon this, aiming to encompass the embodied emissions of manufacturing computing hardware as well as those produced via the inference process (2023). There has also been complementary work measuring the energy used by Transformer models during inference and ways of predicting those costs for different architectures and models (Cao et al., 2021; Ang et al., 2022).

Closest to our work, the HULK benchmark (Zhou et al., 2021) was proposed to measure the relative efficiency-accuracy trade-offs of different pre-trained models, measuring the wall-clock time for different pre-trained models to reach a target accuracy on one of three fine-tuning tasks. Different from Zhou et al. (2021), our work explicitly focuses on the energy and carbon required for fine-tuning (theirs uses time and financial cost as proxies), evaluates a wider variety of fine-tuning tasks and hardware settings in order to elucidate the factors that predict fine-tuning energy requirements, and further contextualizes fine-tuning energy re-

quirements in the bigger picture of ML model life-cycle emissions.

Another related direction of research examines the dynamics of pre-training and fine-tuning of language models and the influence of factors like random seeds and early stopping (Dodge et al., 2020), scaling (Tay et al., 2022) and learning dynamics (Hao et al., 2020). While all of these studies have shed important light on these processes, in practice most of the decisions made remain empirical, with practitioners either referring to previous work (when hyperparameters and other training details are reported), or using techniques such as grid or random search (Bergstra and Bengio, 2012) to converge on optimal parameter values. Our own work builds upon both of these research directions. We study both the pre-training and fine-tuning process, and our experiments for studying their energy intensity are based on the works cited above.

## 3 Methodology

Full training details can be found in Appendix A.1. We release code for replicating our measurements.[2] We encourage others to run our code on their hardware to add to a repository of measurements across different hardware platforms.

### 3.1 Pre-training BERT

In this work we are interested in measuring the energy consumption of fine-tuning, as it compares to other stages of model use: pre-training and inference. In order to establish a comparable baseline for the energy consumption and carbon emissions of pre-training, we pre-trained a BERT-base model (Devlin et al., 2019) from scratch on the same hardware that we use for fine-tuning (Section 3.3). Although in practice pre-training and fine-tuning are often done separately and on different hardware, we fixed the machine for both sets of experiments in order to aid direct comparability of energy usage measurements. Following Devlin et al. (2019), we pre-train our model on the Book-Corpus (Zhu et al., 2015) and the 2020 version of Wikipedia (Foundation, 2020), both downloaded from HuggingFace Datasets (Lhoest et al., 2021).

Our precise pre-training methodology differs slightly from Devlin et al. (2019): our data necessarily differs slightly because the original training corpus was not released along with the model, and we only use the masked language modeling

---

[2]https://github.com/swangxr/FT-energy.git

(MLM) objective without next sentence prediction (NSP) following Liu et al. (2019), who found that removing NSP did not substantially impact end-task performance.

In order to assess the relative impact of using a more efficient pre-trained BERT variant, we also followed the DistilBERT (Sanh et al., 2019) recipe, performing knowledge distillation on our pre-trained BERT-base model.

## 3.2 Fine-tuning BERT

We evaluate the energy consumption and carbon emissions of the fine-tuning process on the tasks in Table 1. We deliberately chose this selection of end-tasks in order to vary fine-tuning dataset size, task type, and sequence length, while also aligning with tasks commonly used in NLP applications.

| Dataset | Task | Examples | Seq. length | |
| | | | Med. | Batch |
| --- | --- | --- | --- | --- |
| Wiki+Books | MLM | 43M | 128 | 128 |
| RTE | NLI | 6K | 56 | 128 |
| MNLI | NLI | 433K | 37 | 90 |
| SQuAD$_{v1}$ | QA | 98K | 159 | 336 |
| SQuAD$_{v2}$ | QA | 142K | 158 | 336 |
| IMDB | Sent. | 50K | 128 | 128 |
| SST2 | Sent. | 70K | 10 | 40 |
| CoNLL$_{2003}$ | NER | 21K | 15 | 53 |
| CoNLL$_{2012}$ | NER | 143K | 20 | 65 |

Table 1: Pre-training and fine-tuning dataset descriptions. We report two sequence length statistics: **Med**ian tokens per sequence in the dataset, and **Batch**, the average maximum sequence length per batch as predicted by simulated sampling. The latter is included as a more direct predictor of computation cost for dynamic padding.

All models are fine-tuned on the BERT models described in §3.1. For each fine-tuning task, we use typical fine-tuning hyperparameters specific to the task or user-reported hyperparameters on current fine-tuned models on HuggingFace, in order to mimic the common real-world use cases. For each task, we dynamically pad sequences to the maximum length in each batch. All fine-tuning hyperparameters are reported in Appendix A.1.

We also report average per-example energy use for inference. All the inference tasks are performed on 1 GPU with batch size 1 on the same machines.

## 3.3 Hardware Platforms

To ensure reproducibility and measure variability across hardware platforms, we replicate experiments across two hardware platforms: One **A100** machine and one **RTX 8000** machine, where each

machine had four GPUs. Pre-training experiments used all 4 GPUs in each machine.

All fine-tuning tasks were performed on the same machines, but using only one GPU. This reflects the typical scenario where fine-tuning is done on a single GPU even if the machine itself has more GPUs. To better compare the energy usage results across pre-training and fine-tuning, we also report an energy usage estimate for BERT-base pre-training on 1 RTX 8000 GPU with hyperparameters equivalent to training on 4 GPUs, extrapolated from a 12-hour partial training run. Details are recorded in Appendix A.1.

## 3.4 Measuring Energy and Carbon

To measure the energy consumed, we use the software tool CodeCarbon (Schmidt et al., 2021). Recent work has found that the existing libraries and code for estimating the carbon emissions of NLP techniques vary in terms of their accuracy and generalizability to different types of hardware (Bannour et al., 2021). To compensate for this, we calibrate the programmatic energy usage readings with a physical energy meter, with which we record energy readings directly from the compute node during experiments. Subsequently, we calculate a coefficient of expected power loss, 1.059, as the average proportion (over runs across fine-tuning tasks) of physical energy reading vs. programmatic energy measurement. Full results are given in Appendix A.2. Thus, the energy consumed in kWh, denoted as $E$, is determined via the formula:

$$E \text{ (kWh)} = 1.059 \cdot \text{codecarbon kWh}$$

Converting the power loss adjusted values to $CO_2$ emissions is done through CodeCarbon using a coefficient specific to the energy grid based on the location of the server from the most recent EPA Power Profiler Data. The conversion factor for the server's location (Pittsburgh, PA) in Table 2 is 1046.1 lbs/MWh, while the factor for the second server's location (Haverford, PA) is 672.8 lbs/MWh. The total kilograms of $CO_2$ emitted, denoted as $C$, is then determined via:

$$C = E \times 1046.1 \frac{\text{lb}}{\text{MWh}} \times \frac{1\text{MWh}}{1000\text{kWh}} \times \frac{1\text{kg}}{2.20462\text{lb}}$$

We convert the $CO_2$ emissions result to human understandable reference values using the EPA Greenhouse Gas Equivalencies Calculator; in Table 2, we also show the equivalent $CO_2$ emissions of miles driven by an average gasoline-powered passenger vehicle.

| Task | Dataset | Training / Fine-tuning | | | | | Inference | |
|------|---------|------------|-----------|------|-----------|-----------|-----------|-----------|
| | | Energy (kWh) | Emissions (kg $CO_2$) | Time | Equiv. # PT runs | Equiv. miles | Energy (kWh) / 1000 ex. | Equiv. # FT runs |
| MLM | 4 GPU (1m @ len 128) | 270.9 | 128.6 | 27 hr | 1.358 | 330 | — | — |
| | 1 GPU (1m @ len 128) | 419.6 | 199.1 | 673 hr | 0.646 | 510 | — | — |
| | 4 GPU (100k @ len 512) | 124.1 | 58.90 | 114 hr | 2.965 | 151 | — | — |
| | Total 4 GPU | 368.0 | 174.6 | 357 hr | 1.000 | 408 | — | — |
| NLI | RTE | 0.008 | 0.004 | 59 s | 45109 | 0.01 | 0.794e-3 | 1k |
| | MNLI | 0.938 | 0.445 | 6700 s | 392 | 1.14 | 7.349e-3 | 127k |
| QA | SQuAD v1 | 0.537 | 0.255 | 3780 s | 685 | 0.65 | 2.157e-3 | 249k |
| | SQuAD v2 | 0.795 | 0.377 | 5604 s | 463 | 0.97 | 2.471e-3 | 322k |
| Sent. | IMDB | 0.151 | 0.072 | 1074 s | 2441 | 0.18 | 0.849e-3 | 178k |
| | SST2 | 0.081 | 0.038 | 587 s | 4540 | 0.10 | 0.691e-3 | 12k |
| NER | CoNLL2003 | 0.021 | 0.010 | 149 s | 17886 | 0.03 | 0.808e-3 | 3k |
| | CoNLL2012 | 0.207 | 0.098 | 1487 s | 1778 | 0.25 | 0.878e-3 | 24k |

Table 2: Energy consumption of pre-training BERT and fine-tuning on the RTX8000 GPU machine. Energy is computed as raw energy measured by CodeCarbon multiplied by a coefficient to correct for power loss (Eq. 3.4). Equiv. miles refers to the approximate number of vehicle miles driven resulting in equivalent $CO_2$ emissions. 1 GPU pre-training costs are extrapolated from a shorter pre-training run lasting only a few hours. Total cost of fine-tuning is derived from 900k steps of pre-training on sequences of length 128 followed by 100k steps on sequences of length 512, following (Devlin et al., 2019). Energy consumption for inference is calculated using single-example batches.

## 4    Results and Discussion

Table 2 shows energy, carbon, and wall-clock time required to fine-tune BERT-base models on the RTX8000 machine. Results on the A100 machine are recorded in Appendix A.3 in Tables 8 and 9.

| Task | Dataset | Energy (kWh) | Time | Emiss. (kg $CO_2$) |
|------|---------|--------------|------|--------------------|
| Distil. | Wiki+Books | 187.74 | 175.5 hr | 89.08 |
| NLI | RTE | 0.004 | 30 s | 0.002 |
| | MNLI | 0.481 | 3447 s | 0.228 |
| QA | SQuAD v1 | 0.276 | 1954 s | 0.131 |
| | SQuAD v2 | 0.412 | 2916 s | 0.196 |
| Sent. | IMDB | 0.077 | 549 s | 0.037 |
| | SST | 0.042 | 306 s | 0.020 |
| NER | CoNLL2003 | 0.011 | 78 s | 0.005 |
| | CoNLL2012 | 0.107 | 762 s | 0.051 |

Table 3: Energy consumption of training (distillation) and fine-tuning DistilBERT on the RTX8000 machine. Energy calculations are the same as in Table 2. Distillation is performed on a pre-trained model, and so the true "total" cost includes the pre-training cost as well.

### 4.1    Pre-training and Distillation

We observe that it requires an additional $50\%$ of the energy cost of pre-training in order to perform knowledge distillation, but it takes nearly $50\%$ less energy to fine-tune on the same tasks using DistilBERT vs. normal BERT (see Table 3). By our estimate, one can fully amortize the up-front cost of distillation within anywhere from 86 fine-tuning

runs of an MNLI-like task, to 47k fine-tunings on an RTE-like task.[3] DistilBERT fine-tuning results on the A100 machine are in Appendix A.3.

### 4.2    Comparing and Predicting Fine-tuning Emissions

We find that, controlling for hardware, energy consumption scales most predictably with *wall clock time* and *number of tokens encountered* during training (including the pad tokens added to sequences to match the maximum sequence length in a batch). The linear relationship between energy consumption and total number of tokens holds similarly on both machines (see Figure 1).Additionally, we observe a consistently higher energy consumption in the RTX 8000 GPU machine. This is likely due to the higher energy overhead and the (in)efficiency of the hardware compared to the A100 GPUs. Other figures in Appendix A.3 illustrate that, in contrast, energy requirements as a function of optimization steps or even number of examples in the dataset can vary significantly across datasets and tasks.

---

[3]Note that cheaper *inference* is often the primary goal of knowledge distillation. Inference is much cheaper than training and therefore requires more to amortize the initial cost of distillation, but inference also occurs much more frequently than training. Models running inference at scale are typically highly optimized with respect to specific deployment settings, so our estimates approximate a lower bound.

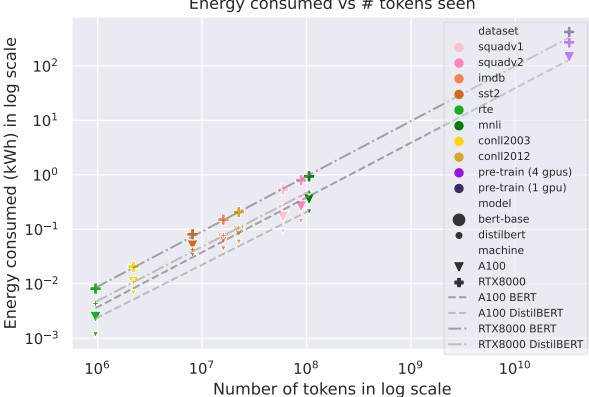

Figure 1: Total energy consumed (kWh) is strongly related to number of tokens seen for BERT models on both A100 and RTX 8000 GPU machines, although the relationship is more predictive on the RTX 8000 machine and energy usage is less consistent with DistilBERT. Note that both axes are in log scale. An alternative view of similar data in Figure 4 distinguishes pre-training workloads' energy consumption slightly from that of fine-tuning tasks.

| Dataset | Seq. length | | kWh / 1k ex. | |
| --- | --- | --- | --- | --- |
| | Med. | Batch | Inf. | FT |
| RTE (NLI) | 56 | 128 | 0.75e-3 | 1.09e-3 |
| MNLI (NLI) | 37 | 90 | 0.70e-3 | 0.80e-3 |
| SQuAD$_{v1}$ (QA) | 159 | 336 | 1.08e-3 | 3.04e-3 |
| SQuAD$_{v2}$ (QA) | 158 | 336 | 1.24e-3 | 3.02e-3 |
| IMDB (Sent.) | 128 | 128 | 0.80e-3 | 1.21e-3 |
| SST2 (Sent.) | 10 | 40 | 0.65e-3 | 0.40e-3 |
| CoNLL$_{2003}$ (NER) | 15 | 53 | 0.76e-3 | 0.49e-3 |
| CoNLL$_{2012}$ (NER) | 20 | 65 | 0.83e-3 | 0.60e-3 |

Table 4: Inference vs. fine-tuning energy requirements across end tasks. We see that fine-tuning energy usage varies according to sequence length much more widely than inference energy usage.

## 4.3 Fine-tuning vs Pre-training

Even for the more reliable predictors of energy consumption and carbon emissions (duration of training and number of tokens processed), the energy cost profiles of pre-training vs. fine-tuning are different, likely due to differences in training infrastructure, training objectives, and sequence lengths typically seen in pre-training vs. fine-tuning (see Figures 4, 3, 5, and 6). Pre-training in general is almost always performed over multiple GPUs which incurs energy costs from communication between GPUs, and often also with gradient accumulation to accommodate large batches. Moreover, sequences are packed together such that batches consist largely or entirely of sequences of identical length equal to the maximum sequence length for the model.

On the other hand, there are many types of fine-tuning tasks where examples consist of sequences of varying lengths significantly shorter than the maximum length that the model has been trained on, as shown in Table 1. Since, effective sequence lengths are determined dynamically at training time (where sequences are padded to the maximum length in each given batch), total training time is not as simple to extrapolate from standard measures of dataset size as in pre-training.

## 4.4 Fine-tuning vs. Inference

Although we do observe that per-example inference efficiency costs are related to sequence lengths, there is overall less variation across datasets and tasks in inference costs compared to fine-tuning costs (see Table 4). This mirrors an observation noted in the HULK benchmark (Zhou et al., 2021), though to the best of our knowledge ours is the first to explicitly draw comparisons across task types and different aspects of dataset size (i.e. number of examples and examples' sequence lengths).

## 4.5 Single vs. Multiple GPUs

In general, typical hardware and data settings for pre-training and fine-tuning tend to differ. Though to the best of our knowledge it is less common to fine-tune causal LMs of this scale on multiple GPUs, we present additional results from multi-GPU fine-tuning on the 4 x RTX8000 machine with the same fine-tuning tasks in Table 10 in Appendix A.3. Our recommendations from an energy efficiency standpoint align with common rules of thumb for effective utilization of hardware; if the resources would be idle otherwise, one could reasonably consider increasing batch size and learning rate to saturate the available hardware for both time- and energy-efficient training.

## 5 Conclusion

We share a procedure for rigorous measurement of energy consumption from causal LM fine-tuning given multiple concrete hardware settings. We hope our work is useful to researchers and practitioners who are interested in obtaining measurements for their own specific hardware, gaining intuitions about factors affecting relative energy costs of different types of fine-tuning tasks, or understanding these energy costs in context of the model lifecycle.

## Limitations

While our work provides important first steps towards a clearer understanding of model fine-tuning's impact on the environment, we note that our experimentation is limited to various token classification, sequence classification, and question answering tasks with BERT-base and DistilBERT-base models. We do not make claims or extrapolations about much larger language models, or models with different architectures, as well as other types of tasks such as summarization. Future work in this direction can expand the number of tasks that we consider as well as feature different architectures such as RoBERTa (Liu et al., 2019).

Additionally, the on-premises hardware infrastructure used for our experimentation is realistic and typical of compute resources in academic settings, but we provide no firsthand evidence of fine-tuning emissions profiles expected from either local model training (where the impracticality of pre-training makes direct comparisons with fine-tuning emissions difficult) or fine-tuning on hardware that is part of much larger scale infrastructure such as on a public cloud. Furthermore, we expect that use of specialized hardware such as TPUs (as opposed to GPUs, which we use) would be associated with different emissions profiles.

## Ethics Statement

Training and deploying ML models and systems requires large amounts of energy, the production of which results in the emission of greenhouse gases. While the goal of our research is to contribute towards a better understanding of the factors that influence these emissions, by carrying out our experiments, we were ourselves responsible for the emission of 350 kg of carbon equivalents. We release our code and the data generated by our experiments to maximize the transparency and reproducibility of our work.

## Acknowledgements

We are grateful to Victor Sanh for his time and patience in answering questions about BERT and DistilBERT pre-training. We would also like to thank our anonymous reviewers for taking the time to provide helpful feedback.

This work was supported in part by a grant from the National Science Foundation Graduate Research Fellowship Program under Grant No. DGE2140739. Any opinions, findings, and conclusions or recommendations expressed in this material are those of the authors and do not necessarily reflect the views of the sponsors.

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

## A  Appendix

### A.1  Pre-training and Fine-tuning Details

For pre-training BERT with MLM, we mainly follow what is listed in Devlin et al. (2019). The specific hyperparameters used for pre-training are listed in Table 5 and were used on both machines.

| Hyperparameters | |
|---|---|
| maximum steps | 1000000 |
| training batch size per device | 64 |
| evaluation batch size per device | 64 |
| maximum sequence length | 128 |
| learning rate | 1e−4 |
| warmup steps | 10000 |
| weight decay | 0.01 |

Table 5: Hyperparameters used for pre-training BERT

In Table 6, we list the set of hyperparameters used to fine-tune each task. All fine-tuning tasks were run on both machines.

**Hardware details** The **A100** machine is located in Haverford, Pennsylvania, USA, and has 4x NVIDIA A100 GPUs with 40GB GDDR SDRAM, 376GB main memory and 32 Intel Xeon processors. The **RTX 8000** machine is located in Pittsburgh, Pennsylvania, USA, and has 4x NVIDIA Quadro RTX 8000 GPUs with 48GB GDDR SDRAM, 36 Intel Xeon processors and 251GB RAM.

## A.2  Kill-A-Watt Measurements

Energy is lost during the process of transferring energy from a power source to the machine. The coefficient for the loss is acquired through the readings from Kill-A-Watt devices. The device measures the instantaneous Watt reading extracted from the wall and displays on the monitor. The measurements for the A100 machine were recorded only on the single node of the cluster containing it (the RTX8000 machine is not part of a larger cluster). The GPU node is connected to two outlets, and we plug separate Kill-A-Watt devices into both. For each instantaneous reading, we read off of and sum up the readings on both Kill-A-Watt devices. To best compare between the package reading and the wall reading, we read off of the device at the same 15 second interval that CodeCarbon records the energy consumed. To convert each instantaneous Watts readings into Kilowatt-Hours, we follow the formula:

$$\text{kWh} = \frac{15 \text{ seconds}}{3600 \text{ seconds}} \times \frac{\text{watts} \times \text{hours}}{1000}$$

We sum up all the calculations for the entire run of a fine-tuning experiment. Then, we can compare the sum of the wall readings with the Code Carbon energy consumption recording. We divide the wall reading over the package reading to get a coefficient, measuring the more realistic energy used. The setup of an instantaneous reading on the A100 machine is shown in Figure 2. The recordings of the wall readings on the A100 GPU machine are recorded in Table 7.

We ran these wall reading experiments for both machines and obtained a coefficient of 1.09 on the A100 machine, and a coefficient of 1.05 on the RTX machine.

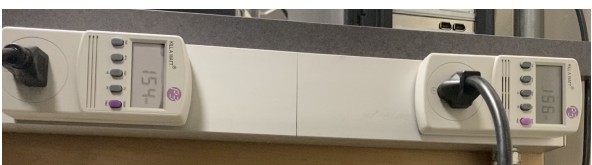

Figure 2: Kill-A-Watt wall reading measurement setup

## A.3  Additional Results

**BERT pre-training and fine-tuning results** Table 8 records the energy consumption results from the A100 GPU machine, located in Haverford, PA. The conversion factor for this location is 672.8 lbs $CO_2$/MWh. We then calculate the emissions using raw output from CodeCarbon, which is listed in Table 8.

**DistilBERT results** Table 9 records the energy consumption and emission on the A100 GPU machine. Distillation is not done on this machine, and all the fine-tunings are done on the DistilBERT trained on the RTX8000 machine. The results on the A100 machine shows a similar trend that fine-tunings on DistilBERT takes around 50% less energy than on BERT base models.

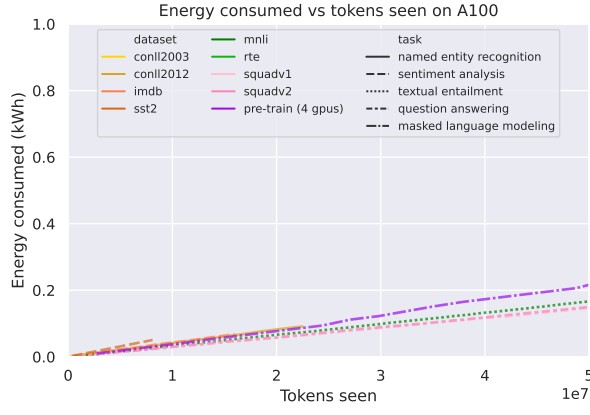

Figure 3: Energy consumed (kWh) against total tokens seen on A100 machine

**Training duration and energy consumption** From Figure 5 and 6, we see that there is a strong correlation between training time and energy consumption, which holds across our models and hardware settings (Figure 7). However, similarly to tokens seen, we observe that pre-training exhibits a slightly different energy consumption profile, as do question answering tasks on the A100 machine (which have the longest sequences out of our fine-tuning tasks). When training time estimates are not feasible in advance (such as in certain hyperparam-

| | RTE | MNLI | IMDB | SST2 | SQuAD v1 | SQuAD v2 | CoNLL 2003 | CoNLL 2012 |
|---|---|---|---|---|---|---|---|---|
| epochs | 3 | 3 | 5 | 3 | 2 | 2 | 3 | 3 |
| train bs | 32 | 32 | 16 | 32 | 32 | 32 | 32 | 32 |
| eval bs | 32 | 32 | 16 | 32 | 32 | 32 | 32 | 32 |
| max seq len | 128 | 128 | 128 | 128 | 384 | 384 | 128 | 128 |
| doc stride | | | | | 128 | 128 | | |
| lr | 2e−5 | 2e−5 | 2e−5 | 2e−5 | 3e−5 | 3e−5 | 5e−5 | 5e−5 |

Table 6: Hyperparameters used for fine-tuning tasks

| Task | Random seed | Wall reading converted (kWh) | Code Carbon reading (kWh) | Power Loss Co-efficient |
|---|---|---|---|---|
| SST2 (no AC) | 42 | 0.103 | 0.088 | 1.170 |
| SST2 | 123 | 0.059 | 0.055 | 1.057 |
| IMDB | 42 | 0.070 | 0.061 | 1.134 |
| IMDB | 42 | 0.079 | 0.071 | 1.122 |
| IMDB | 123 | 0.0793 | 0.073 | 1.092 |
| RTE | 42 | 0.00641 | 0.00602 | 1.065 |

Table 7: Kill-A-Watt experiment recordings for the A100 GPU machine

| Task | Dataset | Energy (kWh) | Emissions (kg $CO_2$) | Training time (s) | Equiv. # PT runs | Equivalent emissions of miles driven |
|---|---|---|---|---|---|---|
| MLM | Wiki + Books | 146.150 | 44.602 | 726553 | 1 | 144 |
| fine-tuning | RTE | 0.002 | 0.002 | 19 | 58442 | 0.005 |
| | MNLI | 0.356 | 0.239 | 2400 | 406 | 0.613 |
| | SQuAD v1 | 0.171 | 0.115 | 1061 | 846 | 0.295 |
| | SQuAD v2 | 0.263 | 0.177 | 1580 | 549 | 0.454 |
| | IMDB | 0.062 | 0.0190 | 478 | 2114 | 0.048 |
| | SST2 | 0.051 | 0.034 | 377 | 2858 | 0.087 |
| | CoNLL2003 | 0.011 | 0.007 | 78 | 13411 | 0.018 |
| | CoNLL2012 | 0.091 | 0.061 | 654 | 1584 | 0.156 |

Table 8: Energy consumption of pre-training BERT and fine-tuning on the A100 GPU machine. Energy is computed as raw energy measured by CodeCarbon multiplied by a coefficient to correct for power loss (Eq. 3.4). Equiv. miles refers to the approximate number of vehicle miles driven resulting in equivalent $CO_2$ emissions. Note that here, unlike in Table 2, the "baseline" pre-training cost is fixed to the cost of 1 million steps of masked language modeling of sequences of length 128.

| Task | Dataset | Energy | Time | Emiss. |
|---|---|---|---|---|
| Entailment | RTE | 0.001 | 10 | 0.001 |
| | MNLI | 0.210 | 1539 | 0.141 |
| QA | SQuAD v1 | 0.091 | 566 | 0.061 |
| | SQuAD v2 | 0.141 | 915 | 0.095 |
| Sentiment | IMDB | 0.045 | 341 | 0.012 |
| | SST | 0.034 | 262 | 0.023 |
| NER | CoNLL2003 | 0.007 | 57 | 0.005 |
| | CoNLL2012 | 0.059 | 448 | 0.040 |

Table 9: Energy consumption of training (distillation) and fine-tuning DistilBERT on the A100 machine. Units and energy calculations are the same as given in Table 8.

eter sweeps), we recommend that researchers and practitioners use token counts estimates (including dynamic padding tokens) if they have reasonable knowledge of their data.

As indicated in Figure 8 and Figure 9, energy increases as the optimization steps increases. This is not surprising given the correlation between training time and energy consumption. However, we see that for different tasks, the energy required for each step is very different. Each step of pre-training takes a longer time, likely due to the higher batch size than all fine-tuning tasks. For QA tasks, the per-step energy consumption is higher than other

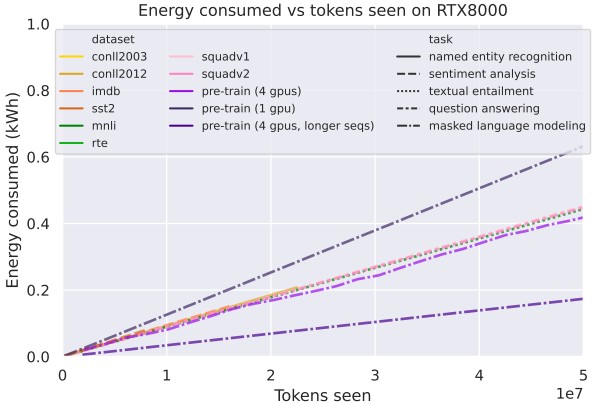

Figure 4: Energy consumed (kWh) against total tokens seen on RTX8000 machine

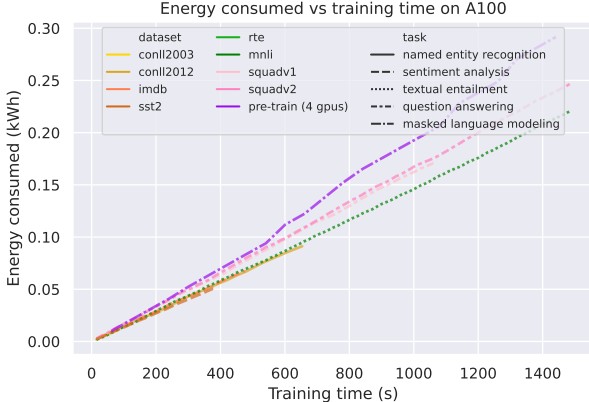

Figure 5: Energy consumed (kWh) against training time (s) on A100 machine, showing the first 30 minutes

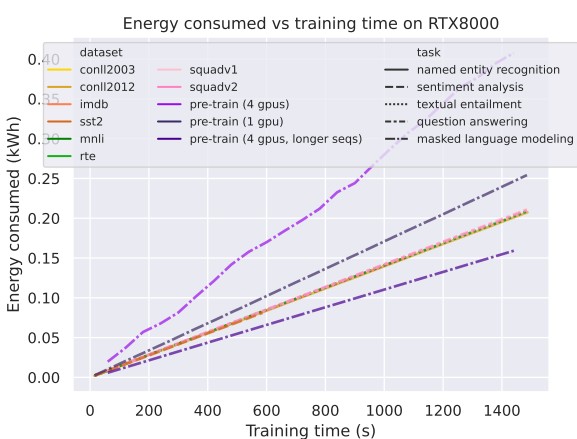

Figure 6: Energy consumed (kWh) against training time (s) on RTX8000 machine, showing the first 30 minutes

tasks. This is likely due to the maximum sequence length of 384 being higher than for the other tasks.

Figure 10 shows the number of examples of the task and the energy consumed in log scale. Similar to Figure 1, we see a direct correlation between

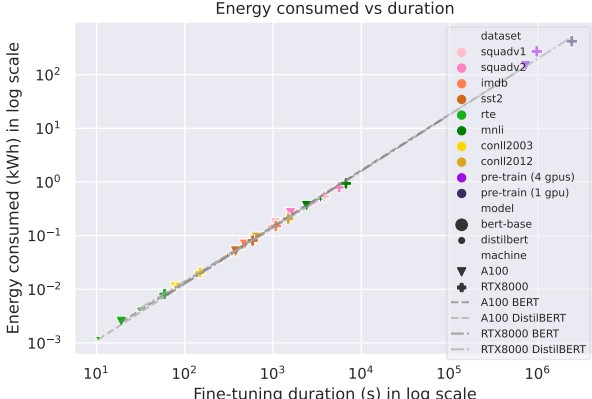

Figure 7: Total training time is strongly predictive of total energy consumed (kWh) for both BERT and DistilBERT models, on both A100 and RTX 8000 GPU machines. Note that both axes are in log scale. Different from token counts, the trendlines themselves are similar as well across models and hardware settings.

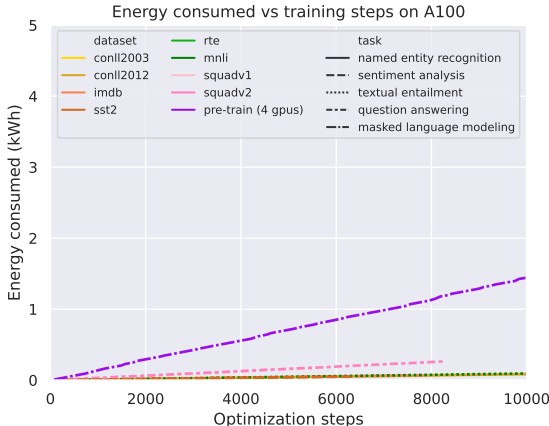

Figure 8: Energy consumed (kWh) against optimization steps on A100 machine machine

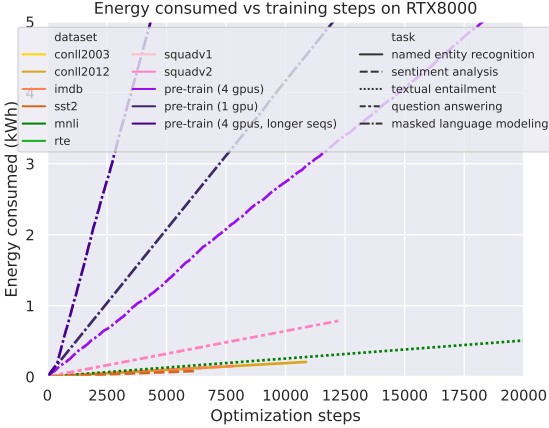

Figure 9: Energy consumed (kWh) against optimization steps on RTX8000 machine

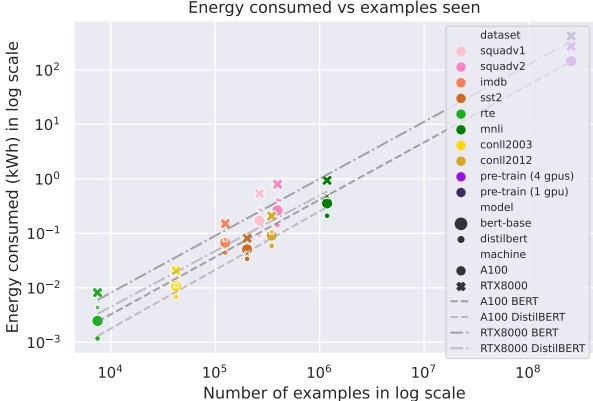

Figure 10: Energy consumed (kWh) against total number of examples seen on both A100 and RTX8000 machines. Comparing with Figure 1, we see that merely counting number of training examples is much less predictive of energy consumption than accounting for example sequence lengths along with batch size (which affects the maximum sequence length in each batch)

higher number of training examples and higher energy usage on both machines.

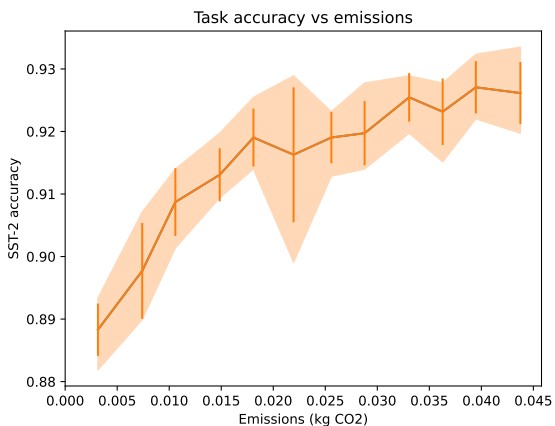

Figure 11: $CO_2$ emissions (kg) against task accuracy for SST-2 fine-tuning on RTX 8000 GPU machine

Figure 11 shows the accuracy of SST2 task as $CO_2$ emissions (kg) increases. As shown previously in Figure 6, energy consumption increases as time increases. Generally, as emission increases, accuracy increases. We can see that as emissions goes up, accuracy is trending towards converging.

**Single- vs. multi-GPU fine-tuning** We fine-tune BERT-base and DistilBERT-base models on the RTX8000 machine as well. If the same hyperparameters as the single-GPU setting are used naively (i.e. the same batch size is split over 4 GPUs), fine-tuning using 4 GPUs can (but does not always) take even longer than using just 1 GPU, and can (but

| Dataset | Duration (s) | | Energy (kWh) | |
|---|---|---|---|---|
| | 1GPU | 4GPU | 1GPU | 4GPU |
| *BERT-base* | | | | |
| RTE (NLI) | 59 | 27 | 8.16e-3 | 8.52e-3 |
| MNLI (NLI) | 6700 | 3014 | 9.38e-1 | 8.64e-1 |
| SQuAD$_{V1}$ (QA) | 3780 | 1356 | 5.37e-1 | 4.28e-1 |
| SQuAD$_{V2}$ (QA) | 5604 | 2011 | 7.95e-1 | 6.16e-1 |
| IMDB (Sent.) | 1074 | 472 | 1.51e-1 | 1.32e-1 |
| SST2 (Sent.) | 587 | 323 | 8.11e-2 | 8.39e-2 |
| CoNLL$_{2003}$ (NER) | 149 | 737 | 2.06e-2 | 2.05e-1 |
| CoNLL$_{2012}$ (NER) | 1487 | 737 | 2.07e-1 | 2.14e-1 |
| *DistilBERT* | | | | |
| RTE (NLI) | 30 | 18 | 4.34e-3 | 4.72e-3 |
| MNLI (NLI) | 3447 | 1760 | 4.81e-1 | 4.98e-1 |
| SQuAD$_{V1}$ (QA) | 1954 | 792 | 2.76e-1 | 2.40e-1 |
| SQuAD$_{V2}$ (QA) | 2916 | 1170 | 4.12e-1 | 3.46e-1 |
| IMDB (Sent.) | 549 | 272 | 7.75e-2 | 7.46e-2 |
| SST2 (Sent.) | 306 | 195 | 4.21e-2 | 5.49e-2 |
| CoNLL$_{2003}$ (NER) | 78 | 436 | 1.10e-2 | 1.18e-1 |
| CoNLL$_{2012}$ (NER) | 762 | 434 | 1.07e-1 | 1.19e-1 |

Table 10: Single-GPU vs. BS and LR-optimized 4-GPU fine-tuning energy requirements across end tasks on the RTX8000 machine, for BERT-base and DistilBERT.

does not always) use about twice as much energy in both BERT-base and DistilBERT. If we increase batch size x 4 with 4 GPUs (and adjust learning rate accordingly), however, and compare single-GPU fine-tuning with multi-GPU fine-tuning (see Table 10), we observe that energy cost is typically similar or even less than when using 1 GPU, while taking around half as much time or less. In both the "naive" and "optimized" multi-GPU settings, the single-vs-multi-GPU difference in energy cost and job duration seems to be related to dataset sequence lengths. Tasks with longer sequences (such as QA tasks, and, to a lesser extent, IMDB and RTE) tend to exhibit more consistent and dramatic energy and time efficiency gains than the other tasks when using 4 GPUs. On the other hand, tasks with shorter sequences (such as NER) tend to require more energy with 4 GPUs, even if the wall-clock efficiency may be improved. One way one might interpret this is a large-enough per-device batch size and typical sequence length is necessary for multi-GPU training to be "worth" the overhead of communication between GPUs.

In light of these observations, our general recommendation is that, if one owns a machine with multiple GPUs, one should consider using all available (idle) GPUs for energy- and time-efficient fine-tuning. Although it is often sufficient to use a single GPU when fine-tuning models of scale similar to ours, and instantaneous energy usage may be

higher using more GPUs, total energy used may end up being less while also requiring less time, especially if the training sequences tend to be longer. On the other hand, tasks with short sequences are likely best kept to a single GPU.