# OpenReview forum: "Energy and Carbon Considerations of Fine-Tuning BERT"
_EMNLP/2023/Conference — EMNLP 2023 Findings_

### Official Review · Reviewer_ikfR · 2023-08-03

**Soundness:** 3

**Ethical Concerns:**

Yes

**Excitement:**

2: Mediocre: This paper makes marginal contributions (vs non-contemporaneous work), so I would rather not see it in the conference.

**Justification For Ethical Concerns:**

To explore the effect of carbon usage on fine-tuning PLMs, this work mimics the whole pre-training and then fine-tuning paradigm and emissions of 350 kg of carbon equivalents.

**Paper Topic And Main Contributions:**

This paper performs a careful empirical study to estimate the carbon usage in the ''pre-train and then fine-tune'' paradigm by reimplementing the BERT pre-training and fine-tuning stage. They conduct experiments to measure carbon usage and provide empirical results.

**Questions For The Authors:**

Question A: If we want to improve the fine-tuning energy efficiency, what can we learn from your paper in addition to some well-known conclusions (e.g., increasing the batch size)?

**Reasons To Accept:**

This paper seems professional in emission estimation and provides a detailed emission result on BERT pre-training and fine-tuning.

**Reasons To Reject:**

1. It is more likely to be a technical report or a blog rather than a research paper: why do we need to consider energy and carbon usage during the fine-tuning stage? What is the difference between pre-training and fine-tuning when considering energy and carbon usage? What is the difference between different fine-tuning strategies?
2. Compared with other learning strategies (e.g., prompt-tuning, in-context learning, low-rank adapter), fine-tuning is more similar to the pre-training stage since they both update all parameters of the LLM. Therefore it will be more valuable to explore carbon usage with learning strategies that do not update the LLM or only update fewer parameters.
3. The organization of this paper can be improved (the conclusion section is missing). It would be better if the authors could move the tables from the appendix to the contents while providing more in-depth discussions (and then extend this paper to a long one).

**Reproducibility:**

4: Could mostly reproduce the results, but there may be some variation because of sample variance or minor variations in their interpretation of the protocol or method.

**Reviewer Confidence:**

3: Pretty sure, but there's a chance I missed something. Although I have a good feel for this area in general, I did not carefully check the paper's details, e.g., the math, experimental design, or novelty.

---

> ### Author Rebuttal · Authors · 2023-08-29
>
> Thank you for taking the time to review our paper. Please see our response below:
>
> > **why do we need to consider energy and carbon usage during the fine-tuning stage?**
>
> We describe differences in assumptions between pre-training and fine-tuning below, but we would first like to address a commonly held belief that fine-tuning generally presents only trivial energy costs. We note in our abstract that, while it is indeed true that a single fine-tuning run’s energy requirements are likely to be negligible compared to pre-training costs, fine-tuning occurs much more frequently and is practiced by many more individuals than pre-training. It is not unrealistic to imagine scenarios where one would run enough fine-tunings (perhaps in a thorough hyperparameter sweep – see Table 2) that the total cost amounts to a sizable fraction of (or even exceeds) the cost of pre-training the model.
>
> Even in our own work, where 1) pre-training was a prominent workload and 2) precise fine-tuning  task accuracy was not a primary objective or consideration (and therefore substantial hyperparameter searches were unnecessary), we estimate 5% of our total emissions from our project are attributable to fine-tuning – equivalent to emissions from over 50 miles driven in a car.
>
> As an aside, we have been involved in and reviewed submissions for previous projects where *only* fine-tuning was performed, where we estimate that the total energy costs easily met or exceeded the 350kg in carbon emissions from our current work. *In each case, the costs arose from reasonable and thorough experimentation across a variety of tasks, models, and methods – and reviewers still asked for additional experiments.*
>
> We believe this is also relevant to the ethical concern raised by the reviewer:
> > **To explore the effect of carbon usage on fine-tuning PLMs, this work mimics the whole pre-training and then fine-tuning paradigm and emissions of 350 kg of carbon equivalents.**
>
> Pre-training, the most energy intensive component of our work, was necessary for contextualizing the fine-tuning measurements on the same hardware. Our hope is that others who hope to contextualize their own fine-tuning costs can cite our findings without having to run the same full experiments themselves.
>
> > **What is the difference between pre-training and fine-tuning when considering energy and carbon usage?**
>
> >  **Compared with other learning strategies (e.g., prompt-tuning, in-context learning, low-rank adapter), fine-tuning is more similar to the pre-training stage since they both update all parameters of the LLM.**
>
> One common concern held by the reviewers was that it was unclear why it is valuable to study energy usage associated with fine-tuning separately from pre-training. We sincerely appreciate the opportunity to clarify – the following is a sentence we ended up cutting from a previous version in order to stay within the page limit, but we see now that it would be beneficial to add back in given an additional page:
> > While this may seem like it should be a straightforward calculation, there are several variables that can influence compute time and energy consumption, ranging from: (1) the type of hardware used for both pre-training and fine-tuning (since this usually differs between the two), (2) the type of task and the type of computation required to carry it out, and (3) intrinsic characteristics of the dataset, such as average sequence length, its similarity with the pre-training dataset, etc.
>
> To elaborate, there are differences in training infrastructure, training objectives, and sequence lengths typically seen in pre-training vs fine-tuning. 1) Pre-training is almost always done using multiple GPUs which incurs additional energy costs from communication between GPUs vs the single GPU setup characteristic of many fine-tuning workloads. 2) Additionally, during pre-training a language modeling objective is used over much longer sequences than typically seen in fine-tuning, which impacts the way that total token count in a dataset factors into energy costs. Sequences are packed together such that batches consist largely or entirely of sequences of identical length equal to the maximum sequence length for the model. 3) On the other hand, there are many different types of fine-tuning tasks where examples consist of sequences of varying lengths significantly shorter than the maximum length that the model has been trained on (see Table 1). Since effective sequence lengths are determined dynamically at training time (where sequences are padded to the maximum length) in each given batch, total training time is not as simple to extrapolate from dataset size and token count as in pre-training.
>
>
> > **What is the difference between different fine-tuning strategies?**
>
> > **it will be more valuable to explore carbon usage with learning strategies that do not update the LLM or only update fewer parameters.**
>
> We agree that investigating other parameter-efficient fine-tuning methods or other learning strategies would be useful follow-up work (although most of the other learning strategies are not common in the masked language models which we experiment with). We focus on dense fine-tuning as a standard and widely practiced method; fine-tuning remains widely used - as can be seen by the almost 300,000 NLP models shared on Hugging Face, the vast majority of which are fine-tuned from a subset of base models - for instance, over 16,000 of them are fine-tuned versions of BERT. Furthermore, just recently Open AI announced that they would be offering a finetuning service for their GPT-series models, indicating that this remains a timely paradigm even in generative settings.
>
> We view our current submission as a first contribution to a line of work investigating the energy demands of task-specific learning across a greater variety of settings. Although there is certainly room for extended follow-up work, we believe that ours is a useful and well-scoped initial finding that should be shared with the community.
>
>
> > **Question A: If we want to improve the fine-tuning energy efficiency, what can we learn from your paper in addition to some well-known conclusions (e.g., increasing the batch size)?**
>
> We agree that improving fine-tuning energy efficiency is a worthy goal, though we do not attempt to propose direct solutions in our current work. Instead, we hope that our work 1) motivates others to consider fine-tuning as a non-trivial workload when planning their experimentation, 2) allows others to estimate fine-tuning costs much more easily than if starting from scratch and 3) encourages others to adopt the practice of estimating the energy/carbon emissions associated with their experimentation, even if each individual fine-tuning is not particularly carbon intensive.
>
>
>
> > **The organization of this paper can be improved (the conclusion section is missing).**
>
> Thank you for this feedback. Given an extra page, we will devote much of it to highlighting specific conclusions from our work.
>
> > **It would be better if the authors could move the tables from the appendix to the contents while providing more in-depth discussions (and then extend this paper to a long one).**
>
> We believe that our work is appropriate for a [short paper](https://2023.emnlp.org/calls/main_conference_papers/#short-papers). Under the types of acceptable papers in the [EMNLP 2023 CFP](https://2023.emnlp.org/calls/main_conference_papers/#contributions), we believe our paper may be considered an "NLP engineering experiment." Historically (e.g. [EMNLP 2021](https://2021.emnlp.org/call-for-papers#short-papers), [ACL 2023](https://2023.aclweb.org/calls/main_conference/)), papers making a "small, focused contribution" using only "a few pages with sufficient level of detail" have been seen as appropriate for submission as a short paper. We are glad that the reviewer found our emissions and energy estimations professional and detailed; we are hopeful that many NLP researchers and practitioners will find our work useful.

---

### Official Review · Reviewer_NXfn · 2023-08-04

**Soundness:** 3

**Excitement:**

2: Mediocre: This paper makes marginal contributions (vs non-contemporaneous work), so I would rather not see it in the conference.

**Paper Topic And Main Contributions:**

In this paper, the authors compare fine-tuning energy use across a suite of standard supervised NLP datasets, including entailment, sentiment analysis, question answering, and NER tasks, with training data sizes ranging from 6K to 400K examples. They also measure energy use across two different sets of hardware. And they suggest increasing batch size for energy-efficient training.


**Reasons To Accept:**

The paper provides a comprehensive analysis of the energy and carbon footprint of both pre-training and fine-tuning BERT models. And it provides a more complete picture of the environmental impact of these widely used NLP models.


**Reasons To Reject:**

The main reason to reject this paper is about the lack of real-world scenarios. Compared to inference, there are very few fine-tuning models, and the author does not delve deeply into the carbon emissions resulting from the inference process.


**Reproducibility:**

4: Could mostly reproduce the results, but there may be some variation because of sample variance or minor variations in their interpretation of the protocol or method.

**Reviewer Confidence:**

3: Pretty sure, but there's a chance I missed something. Although I have a good feel for this area in general, I did not carefully check the paper's details, e.g., the math, experimental design, or novelty.

---

> ### Author Rebuttal · Authors · 2023-08-29
>
> Thank you for the feedback! We are glad you agreed that our paper provides a thorough analysis of energy and carbon intensity of pre-training and fine-tuning BERT models. We are hopeful that our work and its findings will be useful to researchers and practitioners. Please see our response below:
>
> > **Compared to inference, there are very few fine-tuning models**
>
> We respectfully disagree with the reviewer’s statement that there few fine-tuned models – in applied ML settings, fine-tuning remains widely used - as can be seen by the almost 300,000 NLP models shared on Hugging Face, the vast majority of which are fine-tuned from a subset of base models - for instance, over 16,000 of them are fine-tuned versions of BERT. Furthermore, just recently Open AI announced that they would be offering a finetuning service for their GPT-series models, indicating that this remains a timely paradigm even in generative settings.
>
> >  **the author does not delve deeply into the carbon emissions resulting from the inference process**
>
> We include emissions measurements from inference in Table 2 primarily for context – inference efficiency is much more well-studied than fine-tuning, and it is well-known that inference is much less energy-intensive than training. That being said, in the next version of the paper we can more explicitly relate our primary results to the inference measurements:
>
> Although we do observe that per-example inference efficiency costs are related to sequence lengths, there is overall *less variation across datasets and tasks* in inference costs compared to fine-tuning costs. This mirrors an observation noted in the HULK benchmark as well [1], although to the best of our knowledge ours is the first to explicitly draw comparisons across task types and different aspects of dataset size (i.e. number of examples and examples’ sequence lengths).
>
> | Dataset | Task | Median sequence length | Inference energy  (kWh) / 1000 ex. | Fine-tune energy (kWh) / 1000 ex. |
> | --- | --- | --- | --- | --- |
> | RTE | Entailment | 56 | 0.750e-3 | 3.276e-3 |
> | MNLI | Entailment | 37 | 0.704e-3 | 2.388e-3 |
> | SQuAD v1 | Question Answering | 159 | 1.078e-3 | 6.070e-3 |
> | SQuAD v2 | Question Answering | 158 | 1.235e-3| 6.034e-3 |
> | IMDB | Sentiment Analysis | 128 | 0.801e-3 | 6.030e-3 |
> | SST2 | Sentiment Analysis | 10 | 0.653e-3 | 1.204e-3 |
> | CoNLL2003 | Named Entity Recognition | 15 | 0.763e-3 | 1.465e-3 |
> | CoNLL2012 | Named Entity Recognition | 20 | 0.829e-3 | 1.791e-3 |
>
> For the table above, inference was performed on a single RTX8000 with batch sizes of 1. Fine-tuning was performed on the same with batch sizes of 32 except for IMDB where bs=16.
> Note: the authors realized that there were some mistakes in the inference section of Table 2: the result for MNLI was a typo, and we had mistakenly applied a coefficient twice originally. The table above reflects the true values, and we will update the paper accordingly
>
> [1] Xiyou Zhou, Zhiyu Chen, Xiaoyong Jin, and William Yang Wang. 2021. [HULK: An Energy Efficiency Benchmark Platform for Responsible Natural Language Processing](https://aclanthology.org/2021.eacl-demos.39). In *Proceedings of the 16th Conference of the European Chapter of the Association for Computational Linguistics: System Demonstrations*, pages 329–336, Online. Association for Computational Linguistics.

---

### Official Review · Reviewer_9NCQ · 2023-08-05

**Soundness:** 4

**Excitement:**

3: Ambivalent: It has merits (e.g., it reports state-of-the-art results, the idea is nice), but there are key weaknesses (e.g., it describes incremental work), and it can significantly benefit from another round of revision. However, I won't object to accepting it if my co-reviewers champion it.

**Paper Topic And Main Contributions:**

This paper provide an empirical study of the computational costs of pre-training and fine-tuning BERT.

**Questions For The Authors:**

A. Why it is important to measure energy cost of pre-training and fine-tuning separately? Do these two processes have significantly differences when it comes to energy cost?

**Reasons To Accept:**

First, the paper is clearly written and easy to follow, which is appreciable. Second, the goal and claims of the paper are stated clear and supported with ambient empirical evidence. It does provide valuable empirical datapoints for energy cost of fine-tuning BERT that the community can refer to.

**Reasons To Reject:**

The significance of this work might be limited. First, the experiments are only conducted with BERT, which is a relatively out-dated compared to more recent models such as RoBERTa, BART, T5, GPT, etc. If the paper could propose a function to estimate the energy consumed without actually run the experiments, it would be an interesting contribution. Second, the findings like the energy scales with the number of tokens, seem intuitive and not suprising.

**Reproducibility:**

4: Could mostly reproduce the results, but there may be some variation because of sample variance or minor variations in their interpretation of the protocol or method.

**Reviewer Confidence:**

3: Pretty sure, but there's a chance I missed something. Although I have a good feel for this area in general, I did not carefully check the paper's details, e.g., the math, experimental design, or novelty.

---

> ### Author Rebuttal · Authors · 2023-08-29
>
> We thank the reviewer for their thoughtful feedback and suggestions.
>
> > **the experiments are only conducted with BERT, which is a relatively out-dated compared to more recent models such as RoBERTa, BART, T5, GPT, etc.**
>
> Our work focuses on causal language models. We would expect that a study on generative models’ energy considerations would be structured differently such as including much more substantial exploration of inference costs than in our study of BERT-like models. That being said, we would expect our BERT fine-tuning results to be relevant for RoBERTa fine-tuning and inference due to the shared base architecture even though pre-training takes significantly longer with RoBERTa.
>
> > **If the paper could propose a function to estimate the energy consumed without actually run the experiments, it would be an interesting contribution.**
>
> We agree this would be a very valuable contribution! Unfortunately, precise energy usage is ultimately largely hardware-dependent (note the greater deviation from trend lines in the A100 experiments vs the RTX8000 experiments), so proposing a general function that completely obviates the need for empirical measurements is out of scope for this work. That being said, one could extrapolate to predict fine-tuning energy costs of new tasks after running only a couple fine-tunings for calibration. This would be particularly useful in settings where many fine-tuning runs over varied datasets is necessary (e.g. experimenting with different training data recipes in a multi-task learning scenario).
>
>
> > **the findings like the energy scales with the number of tokens, seem intuitive and not suprising**
>
> We agree with the reviewer that our findings generally align with common intuitions, which, until now, were not supported by empirical evidence in a controlled study. That being said, there were multiple *different* alternative conclusions and findings that we were able to empirically rule out as *un*true through our experimentation, that we believe would have been equally intuitive and unsurprising had they turned out to be true instead of the findings presented in the paper. For instance, we would have been unsurprised if using fine-tuning dataset size alone was a sufficient predictor of the amount of energy consumed, perhaps because variation (though we observe this is *not* true in reality; in order to predict the amount of energy consumed in fine-tuning, we must specifically include the per-batch dynamically added padding tokens in the token count total).
>
> To the best of our knowledge, there is no previous work containing our contributions, and our findings were non-trivial to measure empirically. We believe that many researchers and practitioners would find our work useful and interesting.
>
>
>
> > **A. Why it is important to measure energy cost of pre-training and fine-tuning separately? Do these two processes have significantly differences when it comes to energy cost?**
>
> One common concern held by the reviewers was that it was unclear why it is valuable to study energy usage associated with fine-tuning separately from pre-training. We sincerely appreciate the opportunity to clarify – the following is a sentence we ended up cutting from a previous version in order to stay within the page limit, but we see now that it would be beneficial to add back in given an additional page:
>
> > While this may seem like it should be a straightforward calculation, there are several variables that can influence compute time and energy consumption, ranging from: (1) the type of hardware used for both pre-training and fine-tuning (since this usually differs between the two), (2) the type of task and the type of computation required to carry it out, and (3) intrinsic characteristics of the dataset, such as average sequence length, its similarity with the pre-training dataset, etc.
>
> To elaborate further, the energy cost profiles are different, largely due to differences in training infrastructure, training objectives, and sequence lengths typically seen in pre-training vs fine-tuning. 1) Pre-training is almost always done using multiple GPUs which incurs additional energy costs from communication between GPUs vs the single GPU setup characteristic of many fine-tuning workloads. 2) Additionally, during pre-training a language modeling objective is used over much longer sequences than typically seen in fine-tuning, which impacts the way that total token count in a dataset factors into energy costs. Sequences are packed together such that batches consist largely or entirely of sequences of identical length equal to the maximum sequence length for the model. 3) On the other hand, there are many different types of fine-tuning tasks where examples consist of sequences of varying lengths significantly shorter than the maximum length that the model has been trained on (see Table 1). Since effective sequence lengths are determined dynamically at training time (where sequences are padded to the maximum length) in each given batch, total training time is not as simple to extrapolate from standard measures of dataset size as in pre-training.
>
> The reviewers are in agreement that our methodology was described clearly and in detail, and that the claims we make are well-supported. To the best of our knowledge, no previous work provides such detailed estimates of energy costs and carbon emissions associated with fine-tuning BERT models. BERT and DistilBERT are widely used today by researchers and practitioners, and fine-tuning is a common workload that we believe is worth studying separately from pre-training and inference. Overall, we are hopeful that our work will be useful to those within and beyond the NLP community to inform decision-making.

---

### Meta-Review · Area_Chair_m6PH · 2023-09-13

**Recommendation:** 3

**Metareview:**

This paper investigates the energy cost of BERT during the fine-tuning process across various datasets and contextualizes this information in relation to pre-training costs. The reviewers praise the paper for the insightful empirical results and data points it provides on fine-tuning energy use. The reviewers highlight the potential value of these data points as a reference for the community, particularly in the context of hyperparameter optimization and assessing the overall cost associated with fine-tuning BERT models.

However, reviewers have also raised some concerns regarding the paper's limitations, particularly its focus on specific hardware and models. This limitation could restrict the broader applicability of the study's findings as a community reference. While this is an important shortcoming, it might be partially addressed in the next iteration of the paper, e.g., by discussing the observed differences between the two employed GPU variants. For instance, a discussion might provide some guidance to the reader to what extent the empirical results and which aspects of the present study will remain relevant in the future. The next iteration of this paper should also clarify/address the concerns raised by the reviewers regarding the distinction of multi-GPU (for pre-training) and single-GPU (for fine-tuning) results, and the impact of this distinction on the present study.

---

### Decision · Program_Chairs · 2023-10-07

**Decision:**

Accept-Findings

**Comment:**

This paper investigates the energy cost of BERT during the fine-tuning process across various datasets and contextualizes this information in relation to pre-training costs. The reviewers praise the paper for the insightful empirical results and data points it provides on fine-tuning energy use. The reviewers highlight the potential value of these data points as a reference for the community, particularly in the context of hyperparameter optimization and assessing the overall cost associated with fine-tuning BERT models.

However, reviewers have also raised some concerns regarding the paper's limitations, particularly its focus on specific hardware and models. This limitation could restrict the broader applicability of the study's findings as a community reference. While this is an important shortcoming, it might be partially addressed in the next iteration of the paper, e.g., by discussing the observed differences between the two employed GPU variants. For instance, a discussion might provide some guidance to the reader to what extent the empirical results and which aspects of the present study will remain relevant in the future. The next iteration of this paper should also clarify/address the concerns raised by the reviewers regarding the distinction of multi-GPU (for pre-training) and single-GPU (for fine-tuning) results, and the impact of this distinction on the present study.